# *Ecklonia stolonifera* Extract Suppresses Lipid Accumulation by Promoting Lipolysis and Adipose Browning in High-Fat Diet-Induced Obese Male Mice

**DOI:** 10.3390/cells9040871

**Published:** 2020-04-02

**Authors:** Heegu Jin, Kippeum Lee, Sungwoo Chei, Hyun-Ji Oh, Kang-Pyo Lee, Boo-Yong Lee

**Affiliations:** 1Department of Food Science and Biotechnology, College of Life Science, CHA University, Seongnam, Gyeonggi 13488, Korea; heegu94@hanmail.net (H.J.); joy4917@hanmail.net (K.L.); sungwoochei@gmail.com (S.C.); guswl264@naver.com (H.-J.O.); 2Naturalway, Pocheon, Gyeonggi 11160, Korea; kplee@naturalway.co.kr

**Keywords:** *Ecklonia stolonifera* extract, obesity, white adipose tissue, lipolysis, browning

## Abstract

Obesity develops due to an energy imbalance and manifests as the storage of excess triglyceride (TG) in white adipose tissue (WAT). Recent studies have determined that edible natural materials can reduce lipid accumulation and promote browning in WAT. We aimed to determine whether *Ecklonia stolonifera* extract (ESE) would increase the energy expenditure in high-fat diet (HFD)-induced obese mice and 3T3-L1 cells by upregulating lipolysis and browning. ESE is an edible brown marine alga that belongs to the family Laminariaceae and contains dieckol, a phlorotannin. We report that ESE inhibits body mass gain by regulating the expression of proteins involved in adipogenesis and lipogenesis. In addition, ESE activates protein kinase A (PKA) and increases the expression of lipolytic enzymes including adipose triglyceride lipase (ATGL), phosphorylated hormone-sensitive lipase (p-HSL), and monoacylglycerol lipase (MGL) and also thermogenic genes, such as carnitine palmitoyltransferase 1 (CPT1), PR domain-containing 16 (PRDM16), and uncoupling protein 1 (UCP1). These findings indicate that ESE may represent a promising natural means of preventing obesity and obesity-related metabolic diseases.

## 1. Introduction

Obesity is one of the most prevalent global health problems and predisposes toward metabolic diseases such as type 2 diabetes and hyperlipidemia [1,2]. It is caused by excessive fat accumulation when energy intake exceeds energy expenditure [3]. Therefore, lipid metabolism and energy expenditure may be targeted for the treatment or prevention of obesity-associated diseases. Mammals have two types of adipose tissue that have different functions—white adipose tissue (WAT) and brown adipose tissue (BAT) [4]. WAT stores excess energy as triglyceride (TG) and is largely composed of adipocytes containing large unilocular lipid droplets. By contrast, BAT is morphologically and functionally different from WAT, being largely composed of adipocytes containing small multilocular lipid droplets and large numbers of mitochondria. Brown adipocytes express uncoupling protein 1 (UCP1), which is responsible for uncoupling fatty acid oxidation from ATP synthesis, causing the loss of energy as heat [5]. Thus, BAT is a specialized tissue that may represent a useful target for the prevention of obesity [6]. Furthermore, the possibility of converting WAT adipocytes to BAT-like cells has attracted scientific attention, a process referred to as “browning” [7].

Lipolysis—the hydrolysis of TG to release free fatty acids (FFAs) and glycerol—is the pathway for the breakdown of TG stored in WAT [8]. TG consists of three FFAs with a glycerol backbone and is stored in lipid droplets, which are abundant in WAT [9]. Lipolysis is stimulated by the phosphorylation of protein kinase A (PKA), which catalyzes the activation of lipolytic enzymes [10]. In the first step of lipolysis, adipose triglyceride lipase (ATGL) converts TG to diglyceride (DG) and one FFA. In the second step, phosphorylated hormone-sensitive lipase (p-HSL) hydrolyzes DG to produce monoglyceride (MG) and another FFA. Lastly, monoacylglycerol lipase (MGL) hydrolyzes MG and releases a further FFA molecule and glycerol [11]. The released FFAs are transported in the blood and taken up by energy-requiring cells, where mitochondrial fatty acid β-oxidation (FAO) occurs, generating acetyl coenzyme A (acetyl-CoA). This process is also a prerequisite for WAT browning [12]. 

Carnitine palmitoyltransferase 1 (CPT1), a mitochondrial transmembrane enzyme, transports the FFAs into mitochondria, which is the initial and rate-limiting step of FAO [13,14]. FAO is required for UCP1-induced browning [15], during which some of the energy from FAO normally captured as ATP is lost as heat [16]. UCP1 is a mitochondrial inner membrane protein that uncouples electron transport from ATP production, which dissipates the mitochondrial electrochemical gradient and generates heat, which under normal circumstances is used to maintain body temperature [17]. When UCP1 is expressed in larger quantities, this dissipation of the energy stored in FFAs may protect against obesity [18].

*Ecklonia stolonifera* Okamura, which belongs to the family Laminariaceae, is an edible perennial brown marine alga that is widely distributed, at depths of 2–10 m, around the subtidal zone of the southern and eastern coasts of Korea [19] and is rich in polyphenols, including phlorotannins. Dieckol, an indicator component of *Ecklonia stolonifera* extract (ESE), is a major phlorotannin compound that is found only in brown algae [20]. Recent research has shown that it has anti-oxidative, anti-bacterial, anti-inflammatory, anti-hyperlipidemic, and anti-diabetic activities [21,22,23]. Previously, it was also reported that ESE, which contains dieckol, reduces hepatic lipid accumulation in rats with ethanol-induced fatty liver [24]. Moreover, another previous study showed that dieckol has an inhibitory effect on lipid accumulation in 3T3-L1 cells, zebrafish, and mouse models [25]. However, it is not known whether ESE and/or dieckol induce lipolysis and/or adipose browning. Therefore, in the present study, we aimed to determine the effects of ESE on energy expenditure in high-fat diet (HFD)-induced obese mice and the 3T3-L1 cells. Our data suggest that ESE may represent a novel treatment for obesity and the associated metabolic diseases.

## 2. Materials and Methods 

### 2.1. Preparation of *Ecklonia stolonifera* Extract

ESE was obtained from Naturalway Co., Ltd. (Pocheon, Korea). It was prepared from the brown seaweed, *Ecklonia stolonifera*, by 70% edible ethanol extraction for 9 h, filtration, evaporation, and finally freeze-drying at −47 °C. According to previously published high-performance liquid chromatography (HPLC) data [24], the ESE generated contains 23.7 ± 0.21 mg/g dieckol.

### 2.2. Experimental Animals and Diet

The animal studies were in accordance with the criteria outlined in the “Guide for the Care and Use of Laboratory Animals” prepared by the National Academy of Science (NAS) and published by the National Institutes of Health (NIH) and were approved by the Institutional Animal Care and Use Committee of CHA University (IACUC, Approval Number 190193). Male Institute of Cancer Research (ICR) mice (4 weeks of age) purchased from Joong Ah Bio (Suwon, Korea), were housed at 3 mice per cage and maintained in a temperature- and humidity-regulated facility on a 12 h light/dark cycle. After a 1 week period of adaptation, the mice were randomly allocated to four groups (*n* = 9 per group): a chow diet group (CD, containing 10 kcal% as fat; D12450B, Research Diets, NJ, USA), a high-fat diet group (HFD, containing 60 kcal% as fat; D12492, Research Diets), a low-dose ESE-supplemented group (HFD + ESEL), and a high-dose ESE-supplemented group (HFD + ESEH). The mice were orally administered ESE at doses of 50 mg/kg/day (HFD + ESEL) or 150 mg/kg/day (HFD + ESEH) for 6 weeks. During the experimental period, body mass, dietary intake, fasting blood glucose, and rectal temperature were measured each week. At the end of the experiment, the mice were terminally anesthetized with CO_2_ after fasting for 12 h, and then blood and tissue samples were collected.

### 2.3. Body Mass and Food and Water Intake Measurements

During the experimental period, the mice were weighed weekly at 09:00 am using an analytical balance. The food and water intake were calculated each week as the differences between the initial quantities supplied and the amounts of food and water remaining at the end of the week.

### 2.4. Fasting Blood Glucose Measurement

Fasting blood glucose was measured in blood obtained from a tail vein after withholding food for 12 h using an Accu-Check Blood Glucose Meter (Roche, Seoul, Korea).

### 2.5. Rectal Temperature Measurement

The rectal temperatures of the mice were measured weekly using a Testo 925 Type Thermometer (Testo, Lenzkirch, Germany).

### 2.6. Biochemical Analysis

Blood samples were collected at 09:00 am by cardiac puncture under terminal anesthesia. After the blood clotted for 30 min, the serum was separated by centrifugation at 4 °C and 3000× *g* for 20 min. The serum concentrations of triglycerides, total cholesterol, low-density lipoprotein (LDL)-cholesterol, and high-density lipoprotein (HDL)-cholesterol, and the activities of aspartate transaminase (AST) and alanine transaminase (ALT), were determined using colorimetric assay kits (Roche, Basel, Switzerland).

### 2.7. Histological Analysis

Subcutaneous WAT was fixed in 4% paraformaldehyde and embedded in paraffin. Sections were prepared and stained with hematoxylin and eosin (H&E). Photomicrographs were obtained using a Nikon E600 microscope (Nikon, Tokyo, Japan). 

### 2.8. Immunofluorescence

WAT samples were deparaffinized and then incubated with anti-PKA (1:500 dilution), anti-HSL (1:300 dilution), anti-CPT1 (1:200 dilution), or anti-UCP1 (1:200 dilution) antibodies. Secondary anti-mouse fluorescein isothiocyanate (FITC)-conjugated and anti-rabbit Alexa Fluor™ 594-conjugated antibodies were then applied. 4′,6-diamidino-2-phenylindole (DAPI; Thermo Fisher Scientific, Waltham, MA, USA) was used to stain the cell nuclei in the WAT. Slides were mounted with ProLong Gold Antifade reagent (Thermo Fisher Scientific) to maintain fluorescence. Fluorescent images were captured using a Zeiss confocal laser scanning microscope (LSM880; Carl Zeiss, Oberkochen, Germany) and Zen 2012 software (Carl Zeiss).

### 2.9. Cell Culture

3T3-L1 preadipocytes were purchased from the American Type Culture Collection (Manassas, VA, USA) and maintained in Dulbecco’s modified Eagle’s medium (DMEM) containing 10% bovine calf serum (BS) and 1% penicillin/streptomycin (P/S) until 100% confluent. After a further 2 days (D0), this medium was replaced with DMEM containing 10% fetal bovine serum (FBS), 1% P/S, and MDI (0.5 mM 3-isobutyl-1-methylxanthine, 1 μM dexamethasone, and 4 μg/mL insulin). On day 2 (D2), this was replaced with DMEM containing 10% FBS, 1% P/S, and 4 μg/mL insulin, and this medium was replaced every 2 days until day 8 (D8).

### 2.10. Cell Viability

To determine the appropriate concentrations of ESE and dieckol to be used in further investigations, cell viability testing was performed using 3-(4,5-dimethylthiazol-2-yl)-2,5-diphenyltetrazolium bromide (MTT). 3T3-L1 preadipocytes were seeded (5 × 10^2^ cells/well) in 96-well plates and incubated overnight. Stock solutions of ESE and dieckol were prepared in DMSO. Then, the cells were treated with ESE (0, 12, 25, 100, or 200 μg/mL) or dieckol (0, 12, 25, 50, or 100 μM), in comparison to the control cells that had no treatment, for 24 h. After this, 20 μL of a 5 mg/mL MTT solution (yellow color) was added to each well, and the cells were incubated for a further 3 h, and this procedure led to the formation of formazan crystals (purple color). After removing the MTT-containing medium, 100 μL DMSO was added to elute the formazan crystals. Finally, the intensity of the dissolved formazan crystals that are associated with the enzyme activity and the number of viable cells was quantified at 570 nm on a Biotek-enzyme-linked immunosorbent assay (ELISA) reader (BioTek, Winooski, VT, USA).

### 2.11. Oil Red O Staining

After 8 days of differentiation, fully differentiated 3T3-L1 adipocytes were fixed in 4% formaldehyde for 1 h at room temperature and washed twice with 60% isopropanol. The fixed cells were stained with Oil red O (ORO) solution for 30 min and then washed with distilled water. After drying, the stained cells were imaged and the stain was eluted using 100% isopropanol. The absorbance was measured at 490 nm. To analyze hepatic lipid accumulation, cryostat sections of the liver were stained with ORO solution, and photomicrographs were obtained.

### 2.12. Western Blot Analysis

Cells and tissues were lysed in lysis buffer (iNtRON Biotechnology, Seoul, Korea), and the lysate protein concentrations were quantified using a protein assay kit (Bio-Rad, Hercules, CA, USA). Equal amounts of protein were subjected to SDS-PAGE and electrotransferred to membranes. Then, the membranes were blocked with 5% skim milk for 1 h, washed with Tris-buffered saline containing Tween 20 (TBST), incubated with primary antibodies overnight at 4 °C, and then exposed to horseradish peroxidase-conjugated secondary antibodies. Antibodies targeting CCAAT/enhancer-binding protein alpha (C/EBPα), peroxisome proliferator-activated receptor gamma (PPARγ), fatty acid-binding protein 4 (FABP4), sterol regulatory element-binding protein 1 (SREBP1), lysophosphatidic acid acyltransferase theta (LPAATθ), lipin 1, diacylglycerol acyltransferase 1 (DGAT1), phosphorylated protein kinase A (p-PKA, Ser 114), and glyceraldehyde 3-phosphate dehydrogenase (GAPDH) were purchased from Santa Cruz Biotechnology (Dallas, TX, USA); antibodies targeting adipose triglyceride lipase (ATGL) and phosphorylated hormone-sensitive lipase (p-HSL, Ser 563) were purchased from Cell Signaling Technology (Danvers, MA, USA); antibodies targeting monoacylglycerol lipase (MGL), carnitine palmitoyltransferase 1 (CPT1), PR domain-containing 16 (PRDM16), and uncoupling protein 1 (UCP1) were purchased from Abcam (Cambridge, UK); and an antibody targeting β-actin was purchased from ABM (Richmond, BC, Canada).

### 2.13. Statistical Analysis

Data are expressed as means and standard deviations (SDs), and were analyzed by one-way ANOVA, followed by Tukey’s post-hoc test (IBM SPSS Statistics Version 20.0, Chicago, IL, USA). Statistical significance was accepted when *p* < 0.05.

## 3. Results

### 3.1. ESE Prevents the Development of Obesity in HFD-Induced Obese Mice

To determine whether ESE has an anti-obesity effect, we fed mice CD, HFD, or HFD supplemented with 50 mg/kg/day (HFD + ESEL) or 150 mg/kg/day (HFD + ESEH) ESE for 6 weeks. As shown in Figure 1A,B, the body mass of the mice fed an HFD was much higher than that of the mice fed a CD. However, the body masses of the mice fed the HFD and co-administered ESE were dose-dependently reduced. The mice fed the HFD gained 14.1 ± 1.3 g over 6 weeks, whereas the gains in the ESE-treated HFD-fed mice were 9.1 ± 1.9 g for ESEL and 7.3 ± 1.5 g for ESEH. The masses of the WAT, liver, kidney, lung, and spleen were also measured at the end of the experiment, and this showed that ESE-treated HFD-fed mice had smaller WAT depots than the HFD-fed mice, but there were no differences between the groups in the masses of the other organs (Figure 1C,D). In addition, food and water intake were not affected by ESE treatment (Figure 1E). 

As shown in Figure 1F, the fasting blood glucose in the HFD group (133.6 ± 3.7 mg/dL) was higher than that in the CD group (85.8 ± 4.8 mg/dL). However, the fasting glucose was significantly lower in both the ESEL (104.4 ± 5.2 mg/dL) and ESEH (93.4 ± 7.7 mg/dL) groups. In addition, the serum concentrations of triglycerides, total cholesterol, and LDL-cholesterol were lower, and that of HDL-cholesterol was higher, in ESE-treated HFD-fed mice than in mice fed the HFD alone (Table 1). These results indicate that ESE not only suppresses WAT mass gain, but also regulates fasting blood glucose and serum lipid levels.

### 3.2. ESE Inhibits Lipid Accumulation in the WAT of HFD-Induced Obese Mice and 3T3-L1 Cells

To evaluate the effects of ESE and dieckol on lipid accumulation, we measured WAT and 3T3-L1 adipocyte size. As shown in Figure 2A, the WAT adipocytes were larger in HFD-fed mice than in mice fed a CD. However, mice administered ESE had dose-dependently smaller adipocytes than those fed HFD only, such that the cells were similar in size to those in the CD group. Before evaluating the effects in 3T3-L1 cells, we first assessed the cytotoxicity of ESE and dieckol in these cells. As shown in Figure 2B, 200 μg/mL of ESE and 100 μM of dieckol were cytotoxic. Therefore, concentrations of ESE (12, 25, 50, or 100 μg/mL) and dieckol (12, 25, or 50 μM) below these were used in further experiments. To assess lipid accumulation, we fully differentiated 3T3-L1 cells and stained them using ORO solution. Examination of these cells showed that ESE and dieckol reduced the number of lipid droplets (Figure 2C) and lipid accumulation in a dose-dependent manner (Figure 2D). Thus, ESE suppresses the HFD-induced increase in adipocyte size in WAT and lipid droplet accumulation in differentiating 3T3-L1 cells.

### 3.3. ESE Reduces Adipogenesis and Lipogenesis in the WAT of HFD-Induced Obese Mice and 3T3-L1 Cells

To determine the mechanisms of the effects of ESE on adipocyte size and lipid accumulation, we performed western blot analysis. As shown in Figure 3A, HFD-fed mice had higher expression levels of proteins involved in adipogenesis (C/EBPα, PPARγ, and FABP4) than CD-fed mice. However, ESE-treated HFD-fed mice showed a lower expression of these proteins in WAT. Similarly, in 3T3-L1 cells, ESE and dieckol significantly reduced the expression levels in a dose-dependent manner (Figure 3B). In addition, ESE reduced the expression of proteins involved in lipogenesis. As shown in Figure 3C, WAT from the HFD group had higher expression levels of lipogenic proteins (LPAATθ, lipin1, and DGAT1) than that from the CD group, but ESE treatment reduced these. Furthermore, ESE and dieckol significantly reduced the expression of lipogenic proteins (SREBP1, LPAATθ, lipin1, and DGAT1) in 3T3-L1 cells (Figure 3D). Taken together, these results suggest that ESE inhibits adipogenesis and lipogenesis in WAT and 3T3-L1 cells. 

### 3.4. ESE Stimulates Lipolysis in the WAT of HFD-Induced Obese Mice and 3T3-L1 Cells

Lipolysis involves the sequential activity of lipolytic enzymes (ATGL, p-HSL, and MGL), which is induced following PKA activation [26]. Therefore, we determined whether ESE increases lipolysis by upregulating the expression of lipolytic enzymes (ATGL, p-HSL, and MGL) using western blot analysis. As shown in Figure 4A, ESE administration increased the phosphorylation of PKA and the expression levels of the downstream proteins (ATGL, p-HSL, and MGL) fourfold in WAT. Moreover, ESE significantly increased the immunofluorescence staining intensity of PKA and p-HSL in WAT sections, as shown in Figure 4B. Consistent with this, dose-dependent significant increases in the protein expression of lipolytic enzymes were identified in 3T3-L1 cells treated with ESE or dieckol (Figure 4C). These results suggest that ESE increases lipolysis in adipocytes by inducing the activation of PKA and increasing the expression of lipolytic enzymes.

### 3.5. ESE Promotes Browning in the WAT of HFD-Induced Obese Mice and 3T3-L1 Cells

BAT uses FFAs released by lipolysis, and CPT1 is essential for the delivery of FFAs into mitochondria [27]. BAT and browned WAT adipocytes are characterized by UCP1 expression and consume energy by oxidizing FFAs inefficiently in mitochondria; therefore, the change from a WAT- to a BAT-like phenotype is associated with an increase in body temperature [28]. Therefore, we determined the effects of ESE on heat generation by measuring the temperatures of the mice. As shown in Figure 5A, rectal temperature was unaffected by which treatment group the mice were in for the first 2 weeks. However, at the end of the experimental period, the ESE-treated groups had higher rectal temperatures (ESEL: 37.38 ± 0.19 °C; ESEH: 37.86 ± 0.11 °C) than the other groups (CD: 36.20 ± 0.07 °C; HFD: 36.32 ± 0.11 °C). We then performed western blot analysis to measure the expression of the thermogenic proteins CPT1, PRDM16, and UCP1. As shown in Figure 5B, HFD-fed mice expressed lower levels of these proteins than CD-fed mice, but ESE administration increased the expression levels of CPT1, PRDM16, and UCP1. Similarly, CPT1 and UCP1 immunoreactivity were higher in the ESE-treated groups, as shown in Figure 5C. In addition, ESE and dieckol dose-dependently increased the expression of the same proteins in 3T3-L1 cells (Figure 5D). These results suggest that ESE promotes browning by increasing the expression of CPT1 and UCP1 in mitochondria, resulting in a greater loss of energy as heat. 

### 3.6. ESE Reduces Hepatic Lipid Accumulation in the Livers of HFD-Induced Obese Mice

HFD-induced obesity is a risk factor for non-alcoholic fatty liver disease (NAFLD) because it is associated with lipid accumulation in the liver [29]. Therefore, we next determined whether ESE affects hepatic lipid accumulation in HFD-induced obese mice. As shown in Figure 6A, hepatic lipid staining by ORO was dose-dependently reduced by ESE in HFD-fed mice. In addition, the serum AST and ALT activities, which are widely used indicators of liver damage [30], were higher in the HFD than in the CD group, but reduced by ESE treatment (Figure 6B). Furthermore, we performed western blot analysis to measure the expression levels of the key hepatic lipogenic proteins SREBP1, lipin1, and DGAT1 [31]. As shown in Figure 6C, HFD-fed mice expressed higher levels of these hepatic lipogenic proteins in the liver, but ESE treatment significantly reduced these. Taken together, these results suggest that ESE limits hepatic lipid accumulation in HFD-fed mice.

## 4. Discussion

Obesity develops as a result of excessive fat accumulation, which occurs because of an imbalance in energy intake and expenditure. Excess energy is stored as TGs in WAT, which expands through both cellular hyperplasia and hypertrophy [32]. Targeting energy expenditure represents an interesting concept as a means of combating obesity. The discovery of adipose browning and its associated lipid metabolism has generated interest in the search for natural dietary compounds that could be a therapeutic strategy for prevent obesity and its associated metabolic syndrome [33,34]. In the present study, we evaluated two mechanisms whereby ESE might prevent obesity, suppress fat accumulation, and increase energy expenditure—the upregulation of lipolysis and the induction of browning, using HFD-induced obese mice and 3T3-L1 cells, respectively. Our data support the effects of ESE in regulating energy expenditure as a potential new anti-obesity agent. ESE, one of the marine algae, is a natural dietary component that has been used as a food additive or flavoring material in many countries [35]. Marine algae are rich in minerals, dietary fibers, and bioactive compounds, such as polyphenols, which have potential health benefits [36]. Recently, brown marine algae have been identified to be a potential source of anti-obesity substances [37,38,39,40]. It has previously been shown that *Ecklonia stolonifera* has an anti-adipogenic activity in 3T3-L1 adipocytes because it downregulates C/EBPα and PPARγ expression [41,42]. However, the effects of ESE on adipocyte lipolysis and browning has not been investigated. Furthermore, there was a need for more animal studies to assess the efficacy of ESE as an anti-obesity agent. Therefore, in the present study, we assessed the anti-obesity effects of ESE using both in vivo and in vitro models.

The overconsumption of HFD induces body mass gain secondary to the accumulation of WAT [43]. As expected, HFD-fed mice became obese and had higher WAT mass than mice fed a CD. However, the administration of ESE to HFD-fed mice for 6 weeks significantly ameliorated the body mass gain and reduced the size of WAT adipocytes. HFD-induced obesity is also associated with higher concentrations of fasting blood glucose, and serum triglycerides and cholesterol [44]. In the present study, ESE ameliorated the increase in fasting blood glucose, serum triglyceride, total cholesterol, and LDL-cholesterol concentrations, and the reduction in HDL-cholesterol concentration. We also measured the expression of adipogenic proteins in the WAT of each group using western blot analysis. We found that ESE reduced the expression of C/EBPα, PPARγ, and FABP4, which are important for the late stages of preadipocyte differentiation, in HFD-fed mice. Furthermore, ESE reduced the expression of the lipogenic proteins SREBP1, LPAATθ, lipin1, and DGAT1, implying that it may reduce lipid synthesis. The expression of these proteins was also reduced to similar extents by treatment with either ESE or dieckol in 3T3-L1 cells. Therefore, both the in vivo and in vitro data suggest that ESE may prevent the development of HFD-induced obesity by regulating the expression of adipogenic and lipogenic factors. 

HFD-induced obesity is associated with a number of metabolic diseases, including NAFLD, which is characterized by hepatic lipogenesis and progressive steatosis [45]. Hepatic lipid accumulation results from an imbalance between lipid accumulation and disposal, and the key metabolic pathway of lipogenesis are regulated by hepatic lipogenic proteins including SREBP1, lipin1, and DGAT1 [46,47,48]. In the present study, ORO staining showed that the administration of ESE reduces hepatic lipid accumulation and the expression of these lipogenic factors in the liver of HFD-fed mice, which implies that ESE inhibits hepatic lipogenesis. 

Lipolysis is the hydrolysis of TGs stored in intracellular lipid droplets. In adipocytes, the hydrolysis of TGs provides fuel for FAO, and this catabolic process is activated by PKA and involves several lipolytic enzymes (ATGL, HSL, and MGL) [49]. Specifically, a recent study showed that the phosphorylation of HSL is induced following PKA activation, which releases FFAs that can be oxidized in mitochondria [50,51]. In the present study, we found that ESE treatment increased the expression of p-PKA, ATGL, p-HSL, and MGL in HFD-fed mice. Consistent with this, there was significantly higher PKA and p-HSL immunofluorescence in the WAT of ESE-treated groups than in the HFD-only group. In addition, ESE increased the expression of CPT1, an enzyme that facilitates the transfer of FFAs into mitochondria and is rate limiting for FAO. Likewise, ESE and dieckol increased the expression of these lipolytic enzymes and CPT1 in 3T3-L1 cells. Therefore, ESE may increase energy consumption by increasing the expression of these proteins in WAT. 

UCP1 is expressed in the mitochondrial inner membrane and is responsible for the energy-wasting phenotype of BAT. In addition, it has been reported that BAT-like trans-differentiated subcutaneous WAT expresses high levels of UCP1 and shows greater mitochondrial activity [52,53]. In the present study, western blotting showed that ESE increased the expression of thermogenic genes (UCP1 and PRDM16) in WAT from HFD-fed mice and 3T3-L1 cells in a dose-dependent manner. Consistent with this, there were higher intensities of UCP1 and CPT1 immunostaining in WAT from HFD-fed mice. Furthermore, the rectal temperatures of ESE-treated mice were higher than those of the other groups. These results suggest that ESE may increase energy expenditure by increasing the expression of thermogenic genes in the WAT of HFD-induced obese mice. The thermogenic effect of ESE in BAT has not yet been reported, thus further studies are required to understand the details of energy expenditure as heat by UCP1 in BAT, which is morphologically and functionally different from WAT. 

In conclusion, we have shown that ESE inhibits lipid accumulation in WAT by downregulating adipogenesis and lipogenesis. In addition, we have provided evidence that ESE increases lipolysis and FAO, promotes WAT browning, and increases the loss of energy as heat. These findings suggest that ESE has important effects on lipid metabolism and represents a potential dietary means of treating obesity and the related metabolic diseases.

## Figures and Tables

**Figure 1 cells-09-00871-f001:**
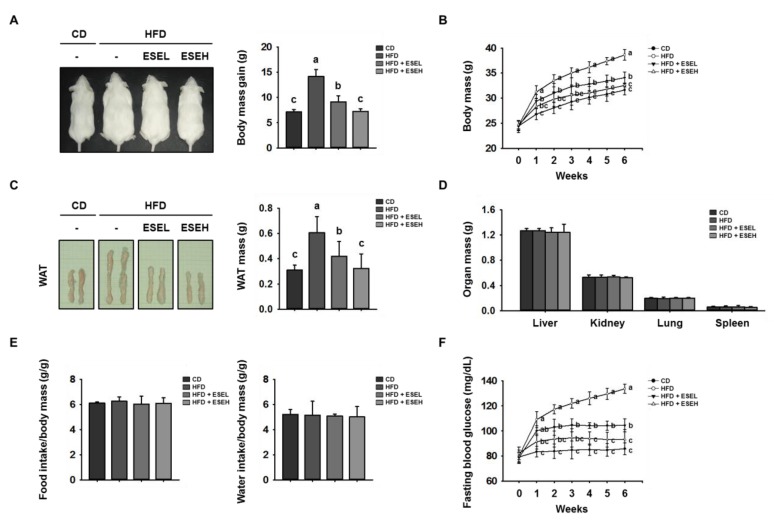
Effects of *Ecklonia stolonifera* extract (ESE) on obesity in high-fat diet (HFD)-induced obese mice. (**A**) Representative images of mice and their body mass gain over 6 weeks. (**B**) Body mass, measured regularly during the 6 weeks of ESE treatment. (**C**) Representative images and the mass of white adipose tissue (WAT) depots. (**D**) Masses of other organs. (**E**) Food and water intake per unit body mass. (**F**) Fasting blood glucose, measured weekly over the 6 weeks of ESE treatment. Data are expressed as means ± SDs (*n* = 8). Values with different letters are significantly different: *p* < 0.05 (a > b > c).

**Figure 2 cells-09-00871-f002:**
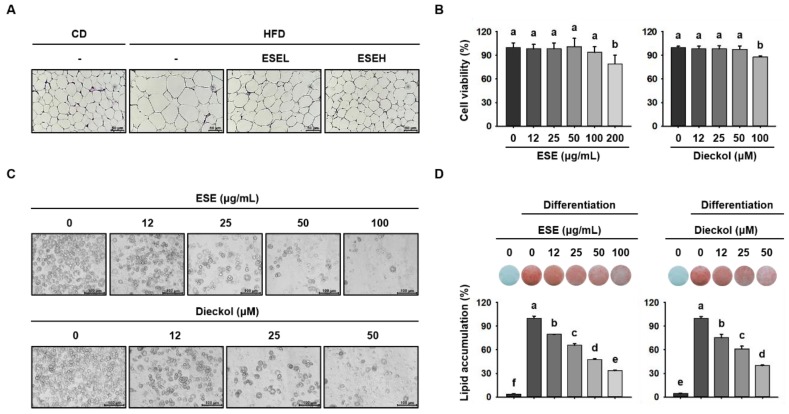
Effects of ESE on adipocyte size and lipid accumulation in HFD-fed mice and 3T3-L1 adipocytes. (**A**) WAT sections stained with hematoxylin and eosin (H&E). Scale bar: 50 μm. (**B**) Viability of 3T3-L1 adipocytes treated with ESE or dieckol for 24 h. (**C**) Microscopic images of 3T3-L1 adipocytes after 8 days of differentiation. Scale bar: 100 μm. (**D**) Oil red O (ORO) staining of 3T3-L1 cells treated with ESE or dieckol after 8 days of differentiation. Data are expressed as means ± SDs (*n* = 4). Values with different letters are significantly different: *p* < 0.05 (a > b > c > d > e > f). ND, non-differentiation.

**Figure 3 cells-09-00871-f003:**
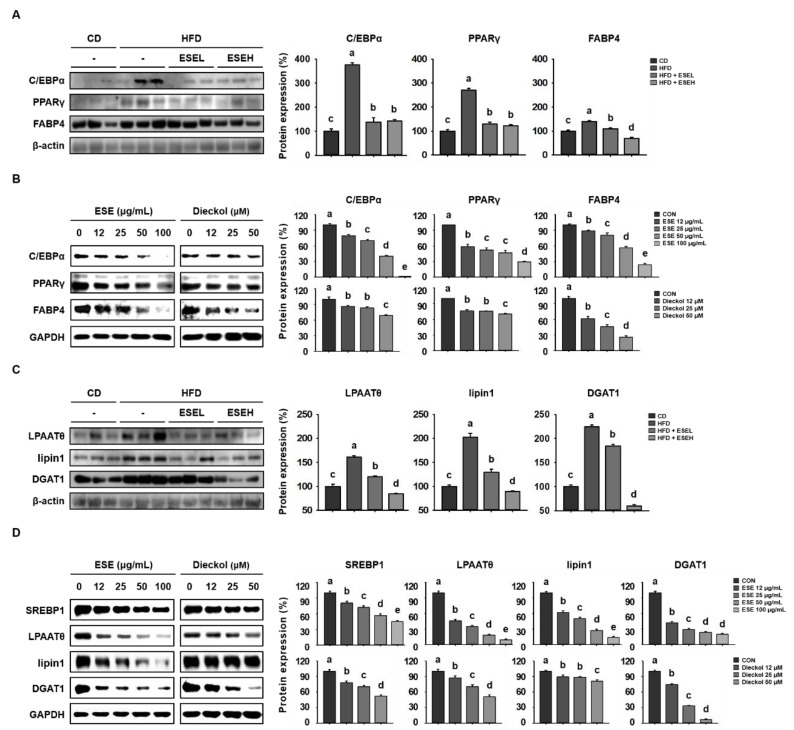
Effects of ESE on the expression of adipogenic and lipogenic proteins in HFD-fed mice and 3T3-L1 adipocytes. (**A**) Western blots of adipogenic proteins (CCAAT/enhancer-binding protein alpha (C/EBPα), peroxisome proliferator-activated receptor gamma (PPARγ), and fatty acid-binding protein 4 (FABP4)) in WAT. (**B**) Western blots of adipogenic proteins in 3T3-L1 cells. (**C**) Western blots of lipogenic proteins (lysophosphatidic acid acyltransferase theta (LPAATθ), lipin1, and diacylglycerol acyltransferase 1 (DGAT1)) in WAT. (**D**) Western blots of lipogenic proteins (sterol regulatory element-binding protein 1 (SREBP1), lysophosphatidic acid acyltransferase theta (LPAATθ), lipin1, and DGAT1) in 3T3-L1 cells. Data are expressed as means ± SDs (n = 4). Values with different letters are significantly different: *p* < 0.05 (a > b > c > d > e).

**Figure 4 cells-09-00871-f004:**
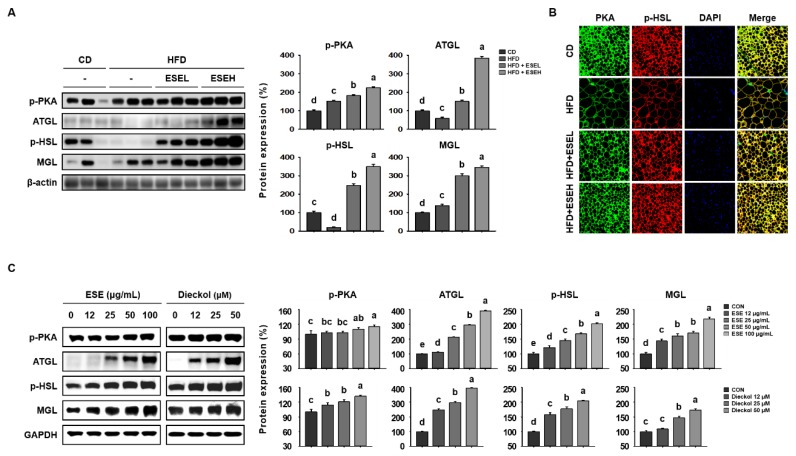
Effects of ESE on the activation of protein kinase A (PKA) and the protein levels of lipolytic enzymes in HFD-fed mice and 3T3-L1 cells. (**A**) Western blots of lipolytic enzymes (phosphorylated protein kinase A (p-PKA), antibodies targeting adipose triglyceride lipase (ATGL), phosphorylated hormone-sensitive lipase (p-HSL), and monoacylglycerol lipase (MGL)) in WAT. (**B**) Immunofluorescence images of WAT (200× magnification). (**C**) Western blots of lipolytic enzymes in 3T3-L1 cells. Data are expressed as means ± SDs (*n* = 4). Values with different letters are significantly different: *p* < 0.05 (a > b > c > d > e).

**Figure 5 cells-09-00871-f005:**
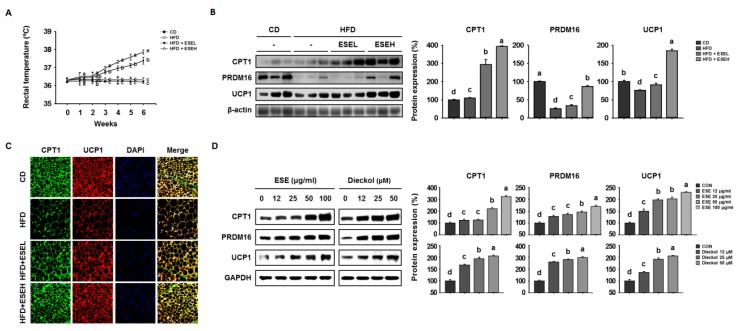
Effects of ESE on thermogenesis and the expression of proteins involved in browning in the WAT of HFD-fed mice and 3T3-L1 cells. (**A**) Rectal temperatures during 6 weeks of ESE treatment. (**B**) Western blots of proteins involved in browning (carnitine palmitoyltransferase 1 (CPT1), PR domain-containing 16 (PRDM16), and uncoupling protein 1 (UCP1)) in WAT. (**C**) Immunofluorescence images of WAT (200× magnification). (**D**) Western blots of proteins involved in browning in 3T3-L1 cells. Data are expressed as means ± SDs (*n* = 4). Values with different letters are significantly different: *p* < 0.05 (a > b > c > d).

**Figure 6 cells-09-00871-f006:**
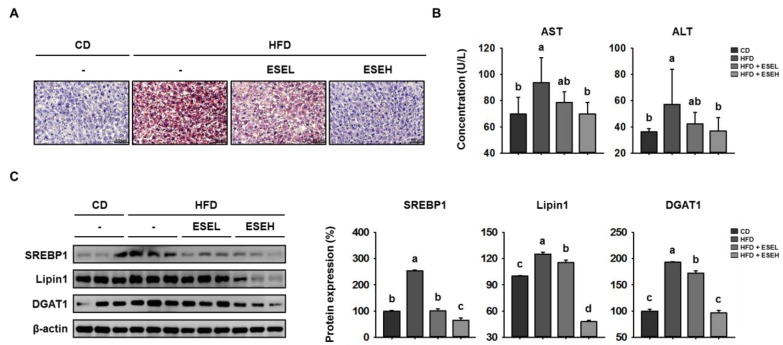
Effects of ESE on hepatic lipid accumulation in HFD-fed mice. (**A**) Oil red O-stained liver sections from mice. Scale bar: 50 μm. (**B**) Serum aspartate transaminase (AST) and alanine transaminase (ALT) activities in HFD-fed mice. (**C**) Western blots of hepatic lipogenic protein expression (SREBP1, Lipin1, and DGAT1). Data are expressed as means ± SDs (*n* = 4). Values with different letters are significantly different: *p* < 0.05 (a > b > c > d).

**Table 1 cells-09-00871-t001:** Effects of ESE administration on blood parameters in HFD-induced obese mice.

	CD	HFD	HFD + ESEL	HFD + ESEH
Triglycerides (mg/dL)	91.4 ± 9.9 ^b^	106.6 ± 12.4 ^a^	99.25 ± 12.2 ^ab^	93.4 ± 6.5 ^b^
Total cholesterol (mg/dL)	154.8 ± 7.6 ^b^	168.5 ± 13.8 ^a^	154.2 ± 12.3 ^b^	160.2 ± 5.8 ^ab^
LDL-cholesterol (mg/dL)	14.8 ± 3.9 ^b^	19.0 ± 1.2 ^a^	17.8 ± 2.4 ^ab^	15.7 ± 0.8 ^b^
HDL-cholesterol (mg/dL)	150.8 ± 2.0 ^a^	105.0 ± 17.2 ^c^	123.5 ± 12.8 ^b^	148 ± 3.1 ^a^

Data are expressed as means ± SDs (*n* = 8). Values with different letters are significantly different: *p* < 0.05 (a > b > c). LDL, low-density lipoprotein. HDL, high-density lipoprotein.

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
