# Peer review of "Ecklonia stolonifera Extract Suppresses Lipid Accumulation by Promoting Lipolysis and Adipose Browning in High-Fat Diet-Induced Obese Male Mice"

_cells, 2020, doi:10.3390/cells9040871_

Round 1

Reviewer 1 Report

The manuscript  entitled “Ecklonia stolonifera Extract suppresses obesity by promoting lipolysis and adipose browning in high-fat diet-induced obese mice” have demonstrated significant results of using Ecklonia stolonifera Extract (ESE) to determine the effects of ESE on energy expenditure in high-fat diet (HFD)-induced obese mice and the 3T3-L1 cells.

The study was written carefully and well in terms of language. However, this work requires minor corrections. In my opinion after correction manuscript should be accepted.

Minor issues:

Page 2, line 69: it should by “cells” not cell line

Page 3, line 133: “…after with 30uL MTT solution was added…” Authors should give concentration of MTT

Page 4, line 137: Author should give information which was the control in MTT assay and how estimated the results. It was cell viability versus the control cultures?

Figure 2: which mean two “0” in figure D description

References:

provide the journals abbreviation

Author Response

Response to Reviewer 1 Comments (Please see the attachment)

We thank you for your time and effort in giving us the opportunity to strengthen our manuscript with your valuable comments. Thus, it is with great pleasure that we resubmit our manuscript for further consideration.

To facilitate your review of our revisions, the following is a point-by-point response to the questions and comments: the original reviewer comments are provided in black color, whereas our answers are given in red. The appropriate changes made in the revised manuscript are highlighted using the “Track Changes” function in Microsoft Word.

Again, thank you for giving us the opportunity to strengthen our manuscript with your valuable comments. We have worked hard to incorporate your feedback and hope that the revised manuscript is suitable for publication in Cells.

Minor issues:

Pont 1: Page 2, line 69: it should by “cells” not cell line

Response 1: Thank you for providing these insights. We agree with you and like your comment, we changed the cell line to “cells” (on page 2, line 73).

Pont 2: Page 3, line 133: “…after with 30uL MTT solution was added…” Authors should give concentration of MTT

Response 2: Thank you for your suggestion. We have reflected this comment by adding a concentration of MTT; 5 mg/mL in the revised manuscript (on page 4, line 141).

Pont 3: Page 4, line 137: Author should give information which was the control in MTT assay and how estimated the results. It was cell viability versus the control cultures?

Response 3: This is an important perspective. We agree with you and have supplemented the 2.10. Cell viability section with explanations of the control in MTT assay and how estimated the result (on page 4, line 140-146). You were correct that cell viability versus the control cultures; 3T3-L1 cells without any treatment (0 µg/mL of ESE or 0 µM of dieckol). Viability in the MTT assay is connected with the quantification of formazan at 570 nm which is linearly associated with the enzyme activity and indirectly the number of viable cells in the culture. High purple color intensity denotes higher cell viability while the decrease in purple color intensity signifies the reduced cell number and thus cytotoxicity of the given substance. Thus, a decreasing absorbance at 570 nm in the 3T3-L1 cells treated with increasing concentration of ESE or dieckol in comparison to the control cells without any treatment. A decreased absorbance in the cells treated with ESE or dieckol suggesting cytotoxicity.

Pont 4: Figure 2: which mean two “0” in figure D description

Response 4: You have raised an important question. We apologize for the confusion about the meaning of terms in Figure 2D. First “0” means the concentration of ESE (0 µg/mL) and dieckol (0 µM) as a control in no differentiation (ND) group of 3T3-L1 cells. Whereas the second “0” means the concentration of ESE (0 µg/mL) and dieckol (0 µM) as a control in the differentiation group. We have included a new term ND, no differentiation to further illustrate Figure 2D (on page 7, line 227). We hope that the edited section clarifies a clearer focus on the meaning of terms.

References:

Pont 5: provide the journals abbreviation

Response 5: Thank you for your suggestion. Note that we have added a list of journal abbreviations (on page 15-18, line 399-532) that were not included in the previous version of the manuscript.

Reviewer 2 Report

Congratulations on your great work, let me indicate certain changes that could improve your presentation.

Abstract and title

a. The title sounds too sharp, it would be better to include something like "seems to suppress".

B. The title should also indicate that these are results exclusively in "induce obese mal mice".

2. Introduction

In the introduction, a more up-to-date bibliography should be used when referring to obesity as the most prevalent health problem.

3. Material and methods

a. It would be interesting to indicate that work guides with animals or that international protocols were followed when ensuring an adequate research methodology with animals, such as the Coucil of Europe agreement or the ARRIVE guidelines.

b. In point 2.3 it should be indicated at what time of day the anthropometry is performed.

c. Likewise, in point 2.6, it would be interesting to know when the analytical sample was taken, and how much time had elapsed since the last intake.

d. The program uses in the statistical analysis is not indicated in point 2.13.

4. Discussion

a. In general, the discussion contains very little discussion of the results of the study, comparing them with other published results. The first paragraph, except for the last line, corresponds rather to the introduction; something similar occurs with lines 315-317 and 343-349. It is recommended to rewrite the discussion, based on the results obtained in the work, which are many and very interesting, and facing them to the published literature. It should be explained what the unexpected or discordant results are attributed, how for example that the CT is greater in HFD+ESSH individuals.

b. It is necessary to include some lines related to the limitations of the study.

c. It would also be interesting to include a paragraph that will assess the research should be directed in the future.

6. References

The bibliography is somewhat outdated, including only 25% of citations published in the last five years.

Author Response

Response to Reviewer 2 Comments (Please see the attachment)

We thank you for your time and effort in giving us the opportunity to strengthen our manuscript with your valuable comments. Thus, it is with great pleasure that we resubmit our manuscript for further consideration.

To facilitate your review of our revisions, the following is a point-by-point response to the questions and comments: the original reviewer comments are provided in black color, whereas our answers are given in red. The appropriate changes made in the revised manuscript are highlighted using the “Track Changes” function in Microsoft Word.

Again, thank you for giving us the opportunity to strengthen our manuscript with your valuable comments. We have worked hard to incorporate your feedback and hope that the revised manuscript is suitable for publication in Cells.

  1. Abstract and title

Pont 1: The title sounds too sharp, it would be better to include something like "seems to suppress".

Response 1: Thank you for providing these insights. This is a valid assessment of our study and we have revised the title; “Ecklonia stolonifera Extract Suppresses Lipid Accumulation by Promoting Lipolysis and Adipose Browning in High-Fat Diet-Induced Obese Male Mice” (on page 1, line 2-4). We have replaced the term “Obesity” into “Lipid Accumulation” to be more appropriate for our paper and hope that you agree.

Pont 2: The title should also indicate that these are results exclusively in "induce obese mal mice".

Response 2: Thank you for your suggestion about the title. We agree with you and have reflected this comment by adding the text “male” to the title (on page 1, line 4) to indicate the exact animal model used in this study.

  1. Introduction

Pont 3: In the introduction, a more up-to-date bibliography should be used when referring to obesity as the most prevalent health problem.

Response 3: We agree with your assessment. We have replaced the references, which published in [1] 2000 and [2] 2003 with more up-to-date citations published in [1] 2017 and [2] 2019 (on page 1, line 29).

[1] González-Muniesa, P.; Mártinez-González, M.-A.; Hu, F.B.; Després, J.-P.; Matsuzawa, Y.; Loos, R.J.F.; Moreno, L.A.; Bray, G.A.; Martinez, J.A. Obesity. Nature Reviews Disease Primers 2017, 3, 17034, doi:10.1038/nrdp.2017.34.

[2] De Lorenzo, A.; Gratteri, S.; Gualtieri, P.; Cammarano, A.; Bertucci, P.; Di Renzo, L. Why primary obesity is a disease? J Transl Med 2019, 17, 169-169, doi:10.1186/s12967-019-1919-y.

  1. Material and methods

Pont 4: It would be interesting to indicate that work guides with animals or that international protocols were followed when ensuring an adequate research methodology with animals, such as the Coucil of Europe agreement or the ARRIVE guidelines.

Response 4: Thank you for your suggestion. This is a valid assessment of regulations and ethics in animal studies. We have now indicated the conventional guidelines for experimentation with animals, “Guide for the Care and Use of Laboratory Animals” prepared by the National Academy of Science (NAS) and published by the National Institutes of Health (NIH) (on page 2, line 82-84).

Pont 5: In point 2.3 it should be indicated at what time of day the anthropometry is performed.

Response 5: We agree with your suggestion and have included the anthropometry performing time; at 0900 hours (on page 3, line 97) that were not included in the previous version of the manuscript.

Pont 6: Likewise, in point 2.6, it would be interesting to know when the analytical sample was taken, and how much time had elapsed since the last intake.

Response 6: We also agree to that and have added the time that blood samples were taken; at 0900 hours, and the blood clotting time; for 30 min after the final intake of the blood sample (on page 3, line 107-108).

Pont 7: The program uses in the statistical analysis is not indicated in point 2.13.

Response 7: Thank you for your suggestion and we have clarified the program name in the revised version of the manuscript (on page 4, line 175). The program that we used to analyze statistical data is IBM SPSS Statistics Version 20.0, which is the software program, produced by SPSS, an IBM company.

  1. Discussion

Pont 8: In general, the discussion contains very little discussion of the results of the study, comparing them with other published results. The first paragraph, except for the last line, corresponds rather to the introduction; something similar occurs with lines 315-317 and 343-349. It is recommended to rewrite the discussion, based on the results obtained in the work, which are many and very interesting, and facing them to the published literature. It should be explained what the unexpected or discordant results are attributed, how for example that the CT is greater in HFD+ESSH individuals.

Response 8: Thank you for providing these insights. You have raised an important point. We agree with you and we have rewritten the discussion (on page 14, line 313-345) to be more in line with your comments. We have rearranged paragraphs and some sentences that are the explanation of ESE were moved to the 1. Introduction section (on page 2, line 62-70) as you commented. We hope that the edited section clarifies the points we attempted to make.

Pont 9: It is necessary to include some lines related to the limitations of the study.

Response 9: Thank you for your suggestion. We agree with you and have included the limitations of the study that the thermogenic effect of ESE in BAT has not yet been reported (on page 15, line 388-390).

Pont 10: It would also be interesting to include a paragraph that will assess the research should be directed in the future.

Response 10: You have raised an interesting point. Please see Response 9 above. We have now indicated the limitations and also the way that the research should be directed in the future (on page 15, line 388-390); further studies are required to understand the details of energy expenditure as heat by UCP1 in BAT, which is morphologically and functionally different from WAT.

  1. References

Pont 11: The bibliography is somewhat outdated, including only 25% of citations published in the last five years.

Response 11:  We agree with you and have incorporated this suggestion throughout our manuscript. We have replaced the outdated references throughout the manuscript with the latest references to use citations published in the last five years as you commented.

References

[1] González-Muniesa, P.; Mártinez-González, M.-A.; Hu, F.B.; Després, J.-P.; Matsuzawa, Y.; Loos, R.J.F.; Moreno, L.A.; Bray, G.A.; Martinez, J.A. Obesity. Nature Reviews Disease Primers 2017, 3, 17034, doi:10.1038/nrdp.2017.34.

[2] De Lorenzo, A.; Gratteri, S.; Gualtieri, P.; Cammarano, A.; Bertucci, P.; Di Renzo, L. Why primary obesity is a disease? J Transl Med 2019, 17, 169-169, doi:10.1186/s12967-019-1919-y.

[3] Romieu, I.; Dossus, L.; Barquera, S.; Blottière, H.M.; Franks, P.W.; Gunter, M.; Hwalla, N.; Hursting, S.D.; Leitzmann, M.; Margetts, B., et al. Energy balance and obesity: what are the main drivers? Cancer Causes Control 2017, 28, 247-258, doi:10.1007/s10552-017-0869-z.

[8] Jocken, J.W.E.; González Hernández, M.A.; Hoebers, N.T.H.; van der Beek, C.M.; Essers, Y.P.G.; Blaak, E.E.; Canfora, E.E. Short-Chain Fatty Acids Differentially Affect Intracellular Lipolysis in a Human White Adipocyte Model. Frontiers in Endocrinology 2018, 8, 372.

[10] Ji, Y.; Lee, J.H.; Han, J.S.; Kong, J.; Kim, J.B. PKA subunit balance plays a key role in lipolysis. The FASEB Journal 2017, 31, 770.715-770.715, doi:10.1096/fasebj.31.1_supplement.770.15.

[13] Qu, Q.; Zeng, F.; Liu, X.; Wang, Q.J.; Deng, F. Fatty acid oxidation and carnitine palmitoyltransferase I: emerging therapeutic targets in cancer. Cell Death Dis 2016, 7, e2226-e2226, doi:10.1038/cddis.2016.132.

[14] Houten, S.M.; Violante, S.; Ventura, F.V.; Wanders, R.J.A. The Biochemistry and Physiology of Mitochondrial Fatty Acid β-Oxidation and Its Genetic Disorders. Annual Review of Physiology 2016, 78, 23-44, doi:10.1146/annurev-physiol-021115-105045.

[16] Demine, S.; Renard, P.; Arnould, T. Mitochondrial Uncoupling: A Key Controller of Biological Processes in Physiology and Diseases. Cells 2019, 8, doi:10.3390/cells8080795.

[17] Solmonson, A.; Mills, E.M. Uncoupling Proteins and the Molecular Mechanisms of Thyroid Thermogenesis. Endocrinology 2016, 157, 455-462, doi:10.1210/en.2015-1803.

[18] Schneider, K.; Valdez, J.; Nguyen, J.; Vawter, M.; Galke, B.; Kurtz, T.W.; Chan, J.Y. Increased Energy Expenditure, Ucp1 Expression, and Resistance to Diet-induced Obesity in Mice Lacking Nuclear Factor-Erythroid-2-related Transcription Factor-2 (Nrf2). The Journal of biological chemistry 2016, 291, 7754-7766, doi:10.1074/jbc.M115.673756.

[19] Manandhar, B.; Wagle, A.; Seong, S.H.; Paudel, P.; Kim, H.R.; Jung, H.A.; Choi, J.S. Phlorotannins with Potential Anti-tyrosinase and Antioxidant Activity Isolated from the Marine Seaweed Ecklonia stolonifera. Antioxidants (Basel, Switzerland) 2019, 8, doi:10.3390/antiox8080240.

[20] Lee, S.; Youn, K.; Kim, D.H.; Ahn, M.-R.; Yoon, E.; Kim, O.-Y.; Jun, M. Anti-Neuroinflammatory Property of Phlorotannins from Ecklonia cava on Aβ(25-35)-Induced Damage in PC12 Cells. Marine drugs 2018, 17, 7, doi:10.3390/md17010007.

[26] Zhang, X.; Zhang, C.C.; Yang, H.; Soni, K.G.; Wang, S.P.; Mitchell, G.A.; Wu, J.W. An Epistatic Interaction between Pnpla2 and Lipe Reveals New Pathways of Adipose Tissue Lipolysis. Cells 2019, 8, doi:10.3390/cells8050395.

[29] Wang, F.; Yang, W.; Xiang, R.; Yuan, J.; Liu, Y.; Chen, K.; Mo, Z. Compound C Protects Mice from Diet-Induced Obesity and Hepatosteatosis. Diabetes 2018, 67, 2019-P, doi:10.2337/db18-2019-P.

[32] Choe, S.S.; Huh, J.Y.; Hwang, I.J.; Kim, J.I.; Kim, J.B. Adipose Tissue Remodeling: Its Role in Energy Metabolism and Metabolic Disorders. Frontiers in endocrinology 2016, 7, 30-30, doi:10.3389/fendo.2016.00030.

[34] Gomez-Zavaglia, A.; Prieto Lage, M.A.; Jimenez-Lopez, C.; Mejuto, J.C.; Simal-Gandara, J. The Potential of Seaweeds as a Source of Functional Ingredients of Prebiotic and Antioxidant Value. Antioxidants (Basel, Switzerland) 2019, 8, 406, doi:10.3390/antiox8090406.

[44] Li, C.-X.; Gao, J.-G.; Wan, X.-Y.; Chen, Y.; Xu, C.-F.; Feng, Z.-M.; Zeng, H.; Lin, Y.-M.; Ma, H.; Xu, P., et al. Allyl isothiocyanate ameliorates lipid accumulation and inflammation in nonalcoholic fatty liver disease via the Sirt1/AMPK and NF-κB signaling pathways. World J Gastroenterol 2019, 25, 5120-5133, doi:10.3748/wjg.v25.i34.5120.

[48] Lazar, I.; Clement, E.; Attane, C.; Muller, C.; Nieto, L. A new role for extracellular vesicles: how small vesicles can feed tumors' big appetite. J Lipid Res 2018, 59, 1793-1804, doi:10.1194/jlr.R083725.

[49] Calderon-Dominguez, M.; Mir, J.F.; Fucho, R.; Weber, M.; Serra, D.; Herrero, L. Fatty acid metabolism and the basis of brown adipose tissue function. Adipocyte 2015, 5, 98-118, doi:10.1080/21623945.2015.1122857.
